# Bi-Allelic *DES* Gene Variants Causing Autosomal Recessive Myofibrillar Myopathies Affecting Both Skeletal Muscles and Cardiac Function

**DOI:** 10.3390/ijms232415906

**Published:** 2022-12-14

**Authors:** Maria Elena Onore, Marco Savarese, Esther Picillo, Luigia Passamano, Vincenzo Nigro, Luisa Politano

**Affiliations:** 1Medical Genetics and Cardiomyology, Department of Precision Medicine, University of Campania “Luigi Vanvitelli”, 80138 Napoli, Italy; 2Folkhälsan Research Center, 00280 Helsinki, Finland; 3Department of Medical and Clinical Genetics, Medicum, University of Helsinki, 00280 Helsinki, Finland; 4Telethon Institute of Genetics and Medicine, 80078 Pozzuoli, Italy; 5Cardiomyology and Medical Genetics, Department of Experimental Medicine, University of Campania “Luigi Vanvitelli”, 80138 Napoli, Italy

**Keywords:** desminopathies, desmin, *DES* gene, cardiomyopathies, heart failure, distal myopathies

## Abstract

Mutations in the human desmin gene (*DES)* may cause both autosomal dominant and recessive cardiomyopathies leading to heart failure, arrhythmias and atrio-ventricular blocks, or progressive myopathies. Cardiac conduction disorders, arrhythmias and cardiomyopathies usually associated with progressive myopathy are the main manifestations of autosomal dominant desminopathies, due to mono-allelic pathogenic variants. The recessive forms, due to bi-allelic variants, are very rare and exhibit variable phenotypes in which premature sudden cardiac death could also occur in the first or second decade of life. We describe a further case of autosomal recessive desminopathy in an Italian boy born of consanguineous parents, who developed progressive myopathy at age 12, and dilated cardiomyopathy four years later and died of intractable heart failure at age 17. Next Generation Sequencing (NGS) analysis identified the homozygous loss-of-function variant c.634C>T; p.Arg212*, which was likely inherited from both parents. Furthermore, we performed a comparison of clinical and genetic results observed in our patient with those of cases so far reported in the literature.

## 1. Introduction

Desminopathy (MIM601419) has been the first-described form of myofibrillar myopathy (MFM), a heterogeneous group of muscle disorders caused by mutations in genes that encode proteins with a main structural or functional role in Z-disc. Desmin is the key component of the intermediate filament (IF) in heart, skeletal and smooth muscles [1]. In skeletal muscle, desmin filaments link the sarcomere to the sarcolemma, extracellular matrix, and nuclear lamina; in addition, desmin binds components of the intermediate filaments. In the cardiac muscle, desmin interacts with the costamere through different IF proteins such as synemin, syncoilin or spectrin. In cardiomyocytes, the Z-discs are surrounded by desmin [2,3].

Desmin has three different domains: a central α-helical domain—called ‘rod’—formed by three continuous α-helical segments, coil 1A, coil 1B and coil 2 connected by two linkers (L1 and L12); a non-α-helical amino-terminal domain representing the ‘head’, and a non-α-helical carboxy-terminal domain representing the ‘tail’ [4]. The amino-terminal domain is necessary for protein assembly while the carboxy-terminal domain is involved in functional protein interactions [5].

Progressive myopathies, cardiac conduction disorders, arrhythmias and cardiomyopathies are the main manifestations of autosomal dominant desminopathies, due to mono-allelic pathogenic variants in *DES* gene. A combination of these clinical features is common.

The pathogenesis of desminopathies is related to the presence of desmin aggregates in the cytoplasm of heart and skeletal muscles. Abnormal protein aggregations and dissolution of myofibrils are the distinction features of desminopathy and, more in general, of myofibrillar myopathies [6].

Clinically, autosomal dominant desminopathies usually present in the third–fourth decade of life with a slowly progressive muscle weakness in distal muscles of lower limbs. Proximal muscle weakness may be observed at later stages. In addition, facial, axial and respiratory muscles are involved, and the patients may present elevated serum creatine kinase (CK) levels. 

Autosomal recessive desminopathies are usually more severe than the dominant forms, with infantile- to adult-onset. The few patients so far reported exhibit variable muscle and cardiac phenotypes including proximal or distal weakness, dilated (DCM), hypertrophic (HCM) or restrictive (RCM) cardiomyopathy. Cardiac death for intractable heart failure in the first and second decade is also described [7].

Here we report a novel case of autosomal recessive desminopathy manifesting as prevalent heart involvement with dilated cardiomyopathy, in a patient who died of intractable heart failure at the age of 17.3. Furthermore, we conduct a review of the literature to compare the clinical and genetic outcomes in patients described so far.

## 2. Patient’s Description

A 12-year-old boy was admitted in May 1989 to the Cardiomyology Service of the Naples University Hospital, for muscle distal weakness. His parents were first cousins. He was the second child of the couple; his two-year-older-sister was healthy. He was born at full-term from spontaneous delivery. A waddling gait was noted at the age of 2.5 by the maternal grandmother. An orthopedist consulted in that period attributed the gait to his flat feet. At the age of six, he was admitted to the Gaslini Hospital in Genoa, due to persisting difficulties in walking and climbing stairs. Muscle needle biopsy performed during hospitalization showed marked variation in fibre size, increased number of nuclear centralizations, few degenerating myofibers and mild increase in endomysial connective tissue, a picture consistent with a dystrophic process. A clear predominance (90%) of type I fibers and presence of microgranuli which appeared colored in red with the Gomori trichrome, were also reported in numerous fibers. Creatin kinase (CK) values were increased about 8x the normal upper limits. The patient was diagnosed with Duchenne muscular dystrophy. However, the molecular analysis of the dystrophin gene with the techinques available at that time did not show any mutation. 

When the patient came under our observation, the physical examination showed elongated facies, ogival palate, waddling gait, difficulties in climbing stairs, positive Gowers manoeuvre, calf pseudo-hypertrophy, and contractures at knees and Achilles tendons. CK values were confirmed to be increased (20.6xN). 

Standard and dynamic ECG showed several ventricular monomorphic ectopic beats. Echocardiography was normal except for an increase in left ventricle diastolic dimensions. Spyrometric parameters (FVC, PEF and FEV1) were within the normal limits. Two years later, cardiac conditions worsened with onset of episodes of ventricular tachycardia (Figure 1A), pulmonary hypertension (Figure 1B) and a progressive deterioration of the ejection fraction consistent with dilated cardiomyopathy. 

Despite the cardio-active therapy with digitalis, diuretics, ACE-inhibitors and anticoagulants, the patient—after an initial improvement—experienced several episodes of acute heart failure. In agreement with his parents, he was included on the waiting list for heart transplant, but he died of cardiac arrest a few months later, at the age of 17.3.

## 3. Results

### 3.1. Gene-Panel Testing

The informed consent for genetic analysis in the patient was obtained from the patient’s father at the time of hospitalization, and from the other family members subsequently. The study was conducted according to the guidelines of the Declaration of Helsinki and approved by the Ethics Committee of the Luigi Vanvitelli University (ID 5586/19 and 8635/19).

Genomic DNA of the patient and family members was extracted from peripheral blood by the salting-out method. A custom gene panel, MotorPlex, was used for the screening of 93 genes considered as genetic causes of primary myopathies that included genes associated to muscular dystrophies, distal myopathies and hereditary cardiomyopathies [8,9]. An in-house pipeline was used to analyze the sequencing-results. Variant annotation was performed using Annovar. Variants were filtered using a minor allele frequency of 0.001 in ExAC and gnomAD databases. Variants’ pathogenicity were assessed according to the ACMG-AMP guidelines. We prioritized those variants predicted as being pathogenic or likely pathogenic.

The investigation showed a homozygous truncating variant located in the coil1B domain of desmin gene [NP_001918, NM_001927.3: c.634C>T; p. Arg212*] (Figure 2). The variant is predicted as likely pathogenic according to the ACMG/AMP guidelines (PVS1 very strong, —PM2 supporting), using the default settings of Varsome (December 2022). The variant is also listed in gnomAD with a very low frequency (f = 0.00001061) and never in homozygosity, and in ClinVar. A recent update reported this variant as likely pathogenic, suggesting its possible link to desminopathies (VCV000201722.8).

No other pathogenic or likely pathogenic variants directly correlated with the observed phenotype were identified. In addition, mutations in the dystrophin and more common genes causing muscular dystrophies such as *LAMA2* and *FKRP* genes had been ruled out. 

### 3.2. Segregation Studies

Sanger sequencing was performed to confirm the presence of the variant in the proband using the following primer pairs [5-GTTTCCACTGCCAGCTTTATCAC-3 (forward) and 5-CTATTCCCAGCCAGAGCCTCAC-3 (reverse)]. Like the father, the sister showed no signs of muscle or heart involvement. It was not possible to analyze the mother’s DNA as she had died before the disease began in her son.

Figure 2 shows the family pedigree and the segregation studies performed using Sanger sequencing. 

## 4. Discussion

MFM disease genes encode proteins that are structural and functional components of the sarcomere, extra-sarcomeric cytoskeleton or protein quality control systems [10,11,12]. In about half of the affected individuals with MFM the genetic defect remains unknown. Desminopathy (MIM601419), Bag3opathy (MIM612954), autosomal recessive epidermolysis bullosa simplex with muscular dystrophy (EBS-MD) (MIM226670) within the group of ‘plectinopathies’ hereditary myopathy with early respiratory failure (HMERF) (MIM603689) within the group of ‘titinopathies’, actin-related MFM, and PYROXD1-myopathy (MIM617258) are associated with early onset of the disease [13]. 

Desmin is an intermediate filament protein mainly expressed in striated and smooth muscles with an important role for structural integrity, mechano-transduction and mechano-sensation.

Initially, desminopathies have been associated with a progressive distal myopathy phenotype starting in the lower legs. However, subsequent studies reported the association between desmin mutations and limb girdle, scapula-peroneal, and generalized myopathy phenotypes [14,15,16].

Cardiac disease manifestations, which may precede, coincide with, or succeed skeletal muscle weakness, comprise true cardiomyopathy as well as various forms of cardiac conduction defects (CCD) and arrhythmias [17,18,19]. Cardiac involvement was correlated neither with the type of DES mutation nor with the severity of skeletal muscle involvement [20].

Nineteen cases of autosomal recessive desminopathies—eleven due to homozygous and eight to compound heterozygous mutations—have been so far reported in the literature. Figure 3 shows the distribution of the mutations along the *DES* gene in these nineteen patients and in our patient.

Table 1 (and references here cited, [7,21,22,23,24,25,26,27,28,29,30,31]) reports the variants (nonsense, small in-frame and out-of-frame deletions, missense variants and canonical splice site mutations) in either homozygosity or compound heterozygosity in patients so far described, as far as their age of onset and clinical presentations.

The analysis of the table shows that most patients are homozygous or compound heterozygous for truncating variants, although four cases show missense variants and four cases a combination of truncating and non-truncating variants.

In all patients, clinical manifestations occurred in the first decade of life. Progressive weakness involving both proximal and distal upper limbs was the first symptom at onset in 8/19 (42%) patients, respiratory distress in 2/19 (10.5%) and ECG abnormalities in 3/19 (15.7%). Symptoms at the onset are not available in four patients. Respiratory insufficiency (severe or chronic) was reported in 5/19 (26.3%) patients, while cardiac involvement was reported in 8/19 (42%) patients, who showed either truncating or non-truncating variants. 

Seven out of the 19 (36.8%) reported patients died, one by respiratory failure at the age of 14, five by cardiac failure or sudden cardiac death at a mean age of 23 (range 17–30 years) and one by respiratory and cardiac complications at 17 years. Recently, a homozygous nonsense variant*-c.700G>T-* has been described in two siblings presenting with a desminopathy at pediatric age [27]. Frequent falling and myopatic facies were the first symptoms. Cardiologic follow-up with ECG and 24-h Holter’s monitoring in both siblings was normal. 

The vast majority of familial desminopathies that follows the autosomal dominant pattern of inheritance is caused by missense mutations leading to single amino acid substitutions. Splice site mutations causing the loss of exon 3 (p.Asp214_Glu245del), small in-frame deletions of one, three or seven codons, and frame shift mutations, which may lead to the expression of truncated desmin protein species, have been reported only in a small number of patients. For the p.Arg350Pro mutation, the most frequently encountered pathogenic desmin missense mutation in Germany, a founder allele has been established [4]. Sporadic cases have also been described. 

Pathogenic variants, located over the entire *DES* gene, have been associated with either dominant or recessive desminopathies, and can affect skeletal muscles (mainly those located in the central alpha-helical rod domain) or cardiac muscle only (mainly those located in the head and tail domains). The coexistence of skeletal muscle and cardiac phenotypes was related to causative variants mainly located in the coil 2 domain. 

The recessive forms, much rarer than the dominant ones, are usually associated with a more severe and more progressive clinical picture. Few variants, in homozygosity or compound heterozygosity were reported in the literature so far, most of which are truncating variants. The greatest number of causative variants occurs in the α-helical rod domain, though two variants of the junction site and a missense variant and a small out-of-frame deletion have also been described, respectively in the tail domain and in the head domain (Figure 3 and Table 1).

Concerning the pathogenic role of some variants compared to others, we still lack an exhaustive understanding of the genotype-phenotype correlation. Truncating variants are, as expected, mostly causative of a recessive phenotype through a loss of function mechanism. This would explain the presence of asymptomatic carriers. The clinical interpretation of rare missense variants is of course more complex. They may cause a dominant disease through a gain of function mechanism, they may cause a recessive disease through loss of function, and some may be hypomorphic variants. This would require further studies.

In Figure 4 we report the survival curve of 20 patients carrying bi-allelic truncating (*n* = 14), bi-allelic non-truncating (*n* = 4) and compound truncating/non-truncating bi-allelic variants (*n* = 2). The figure shows that patients with truncating variants have a longer life expectancy (>40 years), than those with non-truncating variants (30–40 years). It is therefore possible to assume that heterozygotes for truncating variants may be asymptomatic for most of their lives. The small number of patients with mixed bi-allelic variants—truncating/non-truncating—does not allow for any comment.

In this review, the parents of 12/19 patients summarized in Table 1 are reported to be asymptomatic, without clinical evidence of skeletal muscle involvement or cardiac complications. 

In our case, the presence of consanguinity between parents, which supports the condition of homozygosity in the patient and the absence of symptoms in the other members of the family (father and sister) carrying the same mutation in heterozygosity, suggest that the truncating variants are able to express the phenotype only in homozygous conditions.

## 5. Conclusions

Our data confirm that recessive desminopathies are more severe than dominant ones, as patients with loss of function (nonsense or out-of-frame) bi-allelic variants develop clinical manifestations much earlier than dominant forms, in the first decade of life. Furthermore, these variants, which lead to a marked reduction or total absence of the desmin protein, have proven to cause premature sudden cardiac death. 

However, severe cardiorespiratory involvement and premature death can be similarly caused by compound heterozygous or homozygous non-truncating variants (e.g., missense variants or small indels that do not cause frameshift), suggesting that the few cases of recessive forms so far reported do not allow a genotype-phenotype correlation. 

The variant c.634C>T; p. Arg212* described in our patient—albeit not novel—is for the first time reported in homozygous state and associated to a phenotype affecting both skeletal muscles and cardiac function. Furthermore, cardiac involvement was predominant and led to cardiac failure and premature death, though muscle weakness was the first symptom of the disease in the first decade of life. 

## 6. Future Perspective

No specific treatment is available to date for desminopathies. However, a regular neurological, cardiologic and respiratory diagnostic follow-up is recommended to prevent premature death due to cardiac arrhythmias or respiratory failure in these patients. Patients may also benefit from supportive treatment such as pacemakers/implantable cardioverter defibrillators or heart transplantation.

## Figures and Tables

**Figure 1 ijms-23-15906-f001:**
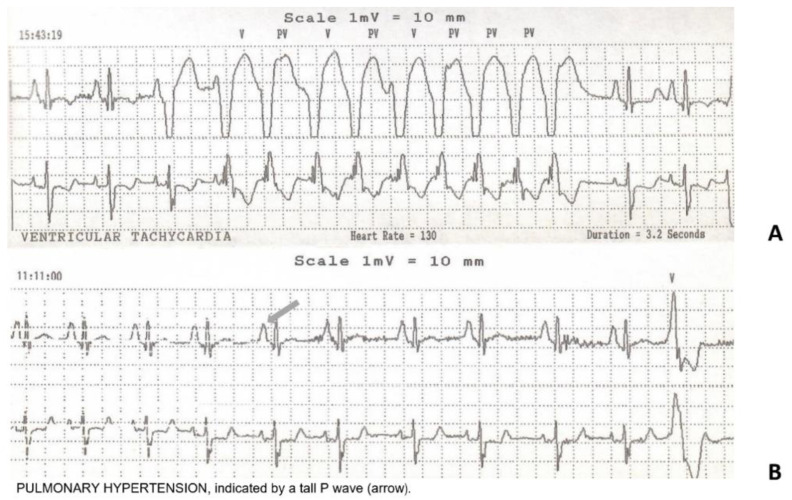
Cardiac Holter tracing. The record shows ventricular tachycardia with a heart rate of 130 bpm (**A**) and pulmonary hypertension highlighted by pulmonary P waves (**B**).

**Figure 2 ijms-23-15906-f002:**
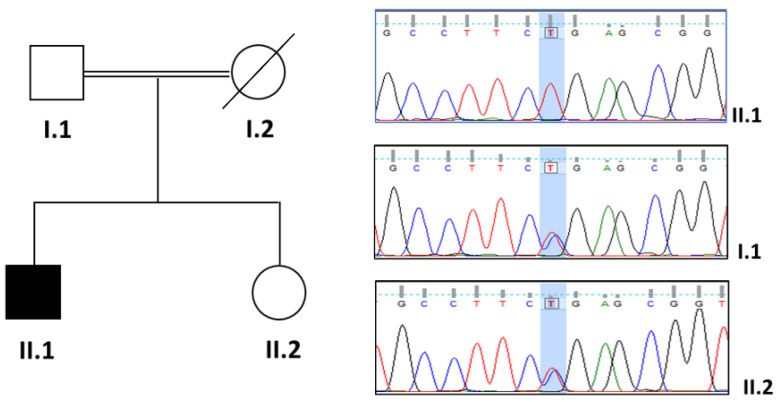
Family pedigree and Sanger analysis. In the family pedigree, a black box indicates the young affected male, born to consanguineous parents. A custom gene panel, MotorPlex, showed a homozygous truncating variant in *DES* gene [NM_001927.3: c.634C>T; p. Arg212*]. Sanger sequencing and segregation studies confirmed the identified DES variant in homozygosity in our patient (II.1) and in heterozygosity in the proband’s father and sister, both healthy (I.1 and II.2). The mother’s DNA was not available.

**Figure 3 ijms-23-15906-f003:**
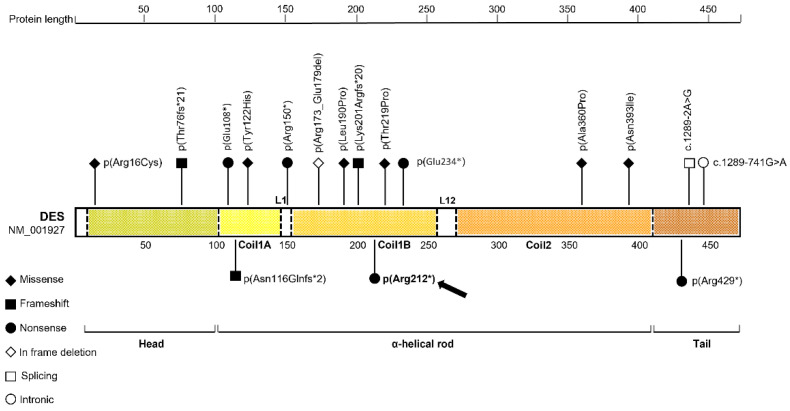
Structural organization of the human desmin and graphical representation of the distribution of bi-allelic variants in *DES* gene. The arrow indicated the variation found in our patient. The different domains and segments of desmin, are painted with distinct patterns. Homozygous and compound heterozygous bi-allelic mutations are aligned according to their location, over the entire *DES* gene. Nonsense, missense, frameshift, deletions, splicing and intronic mutations are reported in the protein graph using different symbols.

**Figure 4 ijms-23-15906-f004:**
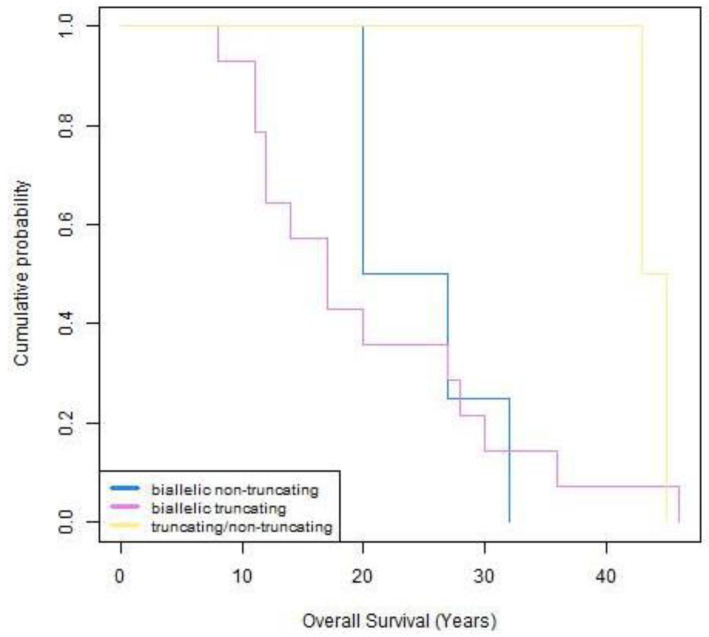
Kaplan–Meier analysis of patients with bi-allelic desmin variants. The curve is based on the patients so far reported in literature, including our patient. The violet line represents patients carrying bi-allelic truncating variants, the light blue line patients carrying bi-allelic non truncating variants while the yellow line patients with bi-allelic truncating/non truncating variants in compound heterozygosity.

**Table 1 ijms-23-15906-t001:** Human recessive desmin mutations and related phenotypes.

Reference	Domain	Zigosity	Gene	Protein	Sex, Age at Onset	First Symptoms	Weakness Distribution	Cardiac Involvement	Respiratory Insufficiency	CK Levels in U/L	Age at Last Follow-Up	Age and Cause of Death
[21]	Head	Hom	*c.46C>T*	p.Arg16Cys	M, n.r.	n.r.	n.r.	RCM, AVB	n.r.	N	32	n.r.
[22]	Coil 1A	Hom	*c.345dupC*	p.Asn116Glnfs*2	M, 3	Facial, neck and upper limb weakness	Generalised weakness and fatigability	None	Chronic	4xN		17-Cardio-respiratory complications
					F, 8	Inability to climb stairs well and run	Generalised weakness and fatigability	None	None	Mildly increased	12	
[23]	Coli 1A	Hom	*c.364T>C*	p.Tyr122His	M, 19	RBBB, BAE	None	RCM	None	588.3	27	
[24]	Linker 1	Hom	*c.448C>T*	p.Arg150*	F, 9	Respiratory distress, LVSD	Predominant distal upper limb weakness	DCM	Severe	347		14-Respiratory insufficiency
[25]	Coil 1B	Hom		p.Arg173_Glu179del	F, 3	Syncopal and cyanotic attacks	Mild diffuse muscle weakness	CMP, VT	None	n.r.		20-Heart failure
[26]	Coil 1B	Hom	*c.655A>C*	p.Thr219Pro	M, 6	ECG anomalies, BVH	Generalised weakness	HCM	None	n.r.		20-Congestive heart failure
[27]	Coil 1B	Hom	*c.700G>T*	p.Glu234*	M, 5; F, 6	Frequent falls, myopathic facies	Generalised weakness; mild muscle atrophy	None	None	700; 762	11; 8	
[28]	Tail	Hom	*c.1289-2A>G*		M, 15	Proximal weakness	Lim-girdle weakness	None	None	117	36	
					F, 27	Proximal weakness	Limb-girdle weakness	None	None	117	46	
Present case	Tail	Hom	*c.634C>T*	p.Arg212*	M, 2.5	Distal weakness	Distal weakness	DCM	None	20.6xN		17.3-Cardiac arrest
[29]	Head	CmHz	*c.226delA*	p.Thr76Fs*21	M, 3	Difficulty in rising from the floor, jumping, running and going upstairs	Predominant proximal weakness	None	None	>1000	12	
	Coil 1A		*c.322G>T*	p.GluG>T	F, 11	Asthma	Less pronounced proximal weakness	None	None	1047	11	
[7]	Coil 1B	CmHz	*c.569T>C*	p.Leu190Pro	F, 20	Progressive weakness	Progressive limb-girdle weakness	LVSD, CMP	Severe	1070	43, 45	
	Coli 2		*c.1289-741G>A*		F, 29							
[30]	Coil 2	CmHz	*c.1078G>C*	p.Ala360Pro	M, 2; M, 9; F, 10	n.r.	Progressive muscle weakness; atrophy	AVB	n.r.	n.r.	20	28,30-conduction system progressive fibrosis
	Coli 2		*c.1187A>T*	p.Asn393Ile								
[31]	Coil 1B	CmHz	*c.600delG*	p.Lys201Argfs*20	F, birth	Hypotonia	Weakness of eye closure; lower facial weakness	DCM	Severe	n.r.	27	
	Tail		*c.1285C>T*	p.Arg429*								

Legenda: CMP = Cardiomyopathy; DCM = Dilated cardiomyopathy; RCM = Restrictive cardiomyopathy; HCM = Hypertrophic Cardiomyopathy; n.r. = not reported; AVB = Atrio-Ventricular Block; VT = Ventricular Tachyarrhythmia; LVSD = Left Ventricular Systolic Dysfunction; RBBB = Right Bundle Branch Block; BAE = Bi-Atrial Enlargement; BVH = BiVentricular Hypertrophy; Hom = Homozygous; CmHz = Compound Heterozygous.

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
