# Peer review of "Bi-Allelic *DES* Gene Variants Causing Autosomal Recessive Myofibrillar Myopathies Affecting Both Skeletal Muscles and Cardiac Function"

_ijms, 2022, doi:10.3390/ijms232415906_

Round 1
Reviewer 1 Report (New Reviewer)
In this study, the authors present a report of a recessive desminopathy case with a novel truncating DES variant and include a useful literature assessment of all previously reported recessive cases.
I'm not sure why the variant is classified using default settings of a generic tool like Varsome? There is a single variant of interest in this study, not a large scale study that might require such automated approaches, and therefore the authors should manually classify the variant according to established clinical genetic approaches. I think the use of the PP5 rule here is problematic, certainly at strong level. This refers to the use of a reputable source to inform classification. The use of such "evidence" has been questioned - see https://www.nature.com/articles/gim201842. The ClinVar entry does include two submissions with P/LP classifications but no evidence is presented to support such assertions. Without this specific evidence, no inferences can be made about pathogenicity. The variant can still be classified as LP with PVS1/PM2 but the lack of any other specific evidence should be noted.
The pathogenicity of the variant seems clear, but any other potentially relevant rare variants detected in the 93 gene panel should be noted.
"The proband’s sister also had the variant, and, like the father, she showed no signs of muscle or heart involvement" - please confirm that the variant is detected in heterozygous form for both the sister and father. Do the authors have data on the cause of the mother's death and whether it is in any way related to the phenotypes described?
For the literature review, the authors may wish to add a recently published report - PMID:35675837. A Kaplan-Meier survival curve (perhaps comparing male and female patients or biallelic truncating vs biallelic non-truncating) would be a nice addition to this section.
"The presence of consanguinity between the patient’s parents, both healthy, and the absence of symptoms in the sister carrying the same mutation, suggest that these variants are able to express the phenotype only in homozygous conditions." - Are the parents both healthy? The text says the mother died before disease onset in the proband. On a broader point, how do the authors explain that most desminopathy cases are dominant and yet the heterozygous relatives of most recessive patients are healthy? Are there particular genotype-phenotype associations, e.g. mostly missense variants in AD, mostly biallelic truncating in AR, such that heterozygous truncating variants may be benign (or less pathogenic)? This warrants more discussion.
Author Response
Please see the attachment.

Reviewer 2 Report (New Reviewer)
Maria Elena Onore et al. report on Bi-allelic DES gene variants causing autosomal recessive desminopathies affecting both skeletal muscles and cardiac function
They reported novel variants.
The clinical and molecular data are very interesting and could interest the readers of the IJMS. However, there are some comments that need to be addressed.
General comments:
+ Follow nomenclature: https://varnomen.hgvs.org/
*- The reference sequence (NP_) and NM_ should also be added.
**Gene name should be italic. Kindly check.
The disorder is Myopathy, myofibrillar, 1 (OMIM 601419), use the proper name of the disorder.
Title: Kindly shorten the title like this: you identified 1 variant, but you wrote variants? Why?
Bi-allelic DES gene variants causing autosomal recessive Myofibrillar Myopathy type 1 affecting both skeletal muscles and cardiac function
Materials and Methods
Put separate headings for NGS, Sanger, and protein modeling.
IRB approval number should be mentioned.
What NGS was used? WES or WGS?
Review of the literature
Move it either to the introduction or discussion. No need for a seperate review of the literature.
Mention all the mutations identified so far in this gene.
Results
It would be better to add the age, height, weight, and HC of the patients with SD.
How the variants were classified as pathogenic ACMG criteria fulfilled?
The novel mutation in DES seems likely pathogenic. Kindly check.
Mention the primers used.
Include the pedigree and Sanger sequencing reports, and patent images if possible.
Add OMIM ID throughout the MS.
Add the NGS filtration steps.
Discussion
Make a clinical table and compare the features of your patient with already reported cases. Try to perform a genotype-phenotype correlation.
A paragraph on future perspectives should be added. For example:
· Proper genetic counseling, the introduction of newborn screening program, and parenteral diagnosis can significantly reduce the burden of such severe skin disorders. This can be accomplished by prenatal genetic testing for monogenetic disorders (PGT-M). PGT and in vitro fertilization are options for parents wishing to have future pregnancies (PMID: 33613643, PMID: 33804821, PMID: 36406136). Although there is no specific management for these cases, patients are treated with supportive treatment.
Author Response
Please see the attachment

This manuscript is a resubmission of an earlier submission. The following is a list of the peer review reports and author responses from that submission.
Round 1
Reviewer 1 Report
Review ijms_1959651: Onore et al., A novel bi-allelic pathogenic variant in DES gene causing fatal dilated cardiomyopathy. A case report and review of literature:
In this manuscript the authors describe the identification, using an NGS approach, of bi-allelic DES mutations in a young patient suffering from recessive desminopathy, who died at the age of 17 years. The clinical details of the patient are described. Moreover, an overview of the relative small group of desminopathy patients also carrying bi-allelic DES mutations and having comparable phenotypes currently available from the literature is given.
Major comments:
The case presented in this manuscript, as well as those from the literature, are clearly and with significant detail described. Having a literature overview of the existing literature on this specific rare patients group will be appreciated by the general public and will have a clear added value. However, it is recommended to present the (clinical) data from the literature in a different way:
-not summarize the individual cases one by one in the main text, but refer for that to the respective table and in the main text summarize the main (shared) clinical features,
-in addition to that, describe, when appropriate, the differences in phenotypes between all patients (the one presented in this manuscript included).
-moreover, evaluate and discuss whether differences were observed between patients being carrier of comp hetero/homozyg LoF variants vs carriers of comp heteroz LoF/missense variants vs carriers of comp heteroz/homozyg of only missense variants. The authors do touch upon this in their conclusion, but this should be evaluated in more depth.
In addition to the above, the manuscript would also improve when the authors would evaluate, as mentioned for this particular case, whether in cases from the literature data is available on the clinical status of family members that are heterozygous carriers of these mutations. In addition also evaluate whether unrelated heterozygous carriers of the in this manuscript collected DES variants are known from literature and/or databases (i.e. clinvar) and what their respective phenotypes are. And when such analysis would indicate that such carriers are without phenotype and these variants represent a specific group of DES mutation only involved in recessive disease then discuss that and speculate about why.
Other comments:
Introduction:
The introduction is relatively long. Please check which elements are truly relevant for this manuscript and only keep that information in the introduction.
Case report:
Concerning the NGS approach: please indicate (more clearly) which genes were analyzed and whether other putatively causal variants were detected. The authors refer to a customized panel described in Ref 8, however also mention that additional genes *muscle disease gene panels) were analyzed in an adapted approaches, so please include that information.
Discussion:
The authors state that “pathogenic variants are located over the entire DES gene. In particularly, those engaging the central alpha-helical rod domain”. This seems to refer to DES variants in general, however please clarify and elaborate on this.
Related to the above, the discussion section is very short. Above several aspects now not included and thus not discussed should be incorporated.
Author Response
Response to Reviewers
We thank the reviewers for their comments and suggestions that we have carefully considered in the reviewing of our manuscript. Our point- to- point reply to each reviewer is listed below.
Reviewer 1
The case presented in this manuscript, as well as those from the literature, are clearly and with significant detail described. Having a literature overview of the existing literature on this specific rare patients group will be appreciated by the general public and will have a clear added value. However, it is recommended to present the (clinical) data from the literature in a different way:
-not summarize the individual cases one by one in the main text, but refer for that to the respective table and in the main text summarize the main (shared) clinical features
Answer: we have summarized - as suggested - the main clinical features of patients listed in the table, in the main text.
-in addition to that, describe, when appropriate, the differences in phenotypes between all patients (the one presented in this manuscript included).
Answer: As suggested, we have compared – where possible - the phenotypes between all patients
-moreover, evaluate and discuss whether differences were observed between patients being carrier of comp hetero/homozyg LoF variants vs carriers of comp heteroz LoF/missense variants vs carriers of comp heteroz/homozyg of only missense variants. The authors do touch upon this in their conclusion, but this should be evaluated in more depth
Answer: Only 5 out 17 cases reported in the table have a compound/heterozygous variant, 2 of them presenting the same mutation, not allowing a genotype-phenotype correlation. Moreover, the analysis of parent’s data - available for 12/17 cases -shows that all the parents are phenotypically normal and do not show signs of skeletal muscle and/or cardiac involvement, suggesting that the variants are likely to express phenotypically only in the recessive condition.
In addition to the above, the manuscript would also improve when the authors would evaluate, as mentioned for this particular case, whether in cases from the literature data is available on the clinical status of family members that are heterozygous carriers of these mutations. In addition also evaluate whether unrelated heterozygous carriers in this manuscript collected DES variants are known from literature and/or databases (i.e. clinvar) and what their respective phenotypes are. And when such analysis would indicate that such carriers are without phenotype and these variants represent a specific group of DES mutation only involved in recessive disease then discuss that and speculate about why.
Answer: See previous answer for the first part of your comment. As far as the case report, the presence of parent’s consanguinity and the lack of skeletal muscle and cardiac symptoms in the patient’s father and sister both heterozygous for the same variant, supports the suggestion that it is likely to express phenotypically only in the recessive condition.
Introduction: The introduction is relatively long. Please check which elements are truly relevant for this manuscript and only keep that information in the introduction.
Answer: We shortened the introduction as suggested.
Case report: Concerning the NGS approach: please indicate (more clearly) which genes were analyzed and whether other putatively causal variants were detected. The authors refer to a customized panel described in Ref 8, however also mention that additional genes *muscle disease gene panels) were analyzed in an adapted approach, so please include that information.
Answer: As suggested, we have indicated which genes were analyzed and whether other putatively causal variants were detected as following: “A custom NGS- based platform, MotorPlex was used for the screening of 93 genes considered as genetic causes of primary myopathies, including genes associated to muscular dystrophies, distal myopathies and hereditary cardiomyopathies. An in-house pipeline was used to analyze the NGS results. Sanger sequencing was performed to confirm the variant. Mutations in the dystrophin and more common genes causing muscular dystrophies were excluded, including. heterozygous mutations in LAMA2 and FKRP genes.”
Discussion: The authors state that “pathogenic variants are located over the entire DES gene. In particularly, those engaging the central alpha-helical rod domain”. This seems to refer to DES variants in general, however please clarify and elaborate on this.
Answer: We have clarified and elaborate this information: “Pathogenic variants located on the entire DES gene have been associated with either dominant or recessive desminopathies affecting only the skeletal muscle (mainly those located in the central alpha-helical rod domain) or the cardiac muscle (mainly those located in the head and tail domains). The coexistence of skeletal-muscle and cardiac phenotypes has mainly been attributed to causative variants in coil 2 domain. As for the recessive desminopathies, few homozygous and compound heterozygous variants have been described in literature so far. Most of them are truncating variants although missense variants have been also identified. Most causative variants occur in the α-helical rod domain. Two splice site variants are reported in the tail domain while a missense variant and a small out-of-frame deletion occur in the head domain (Figure 2 and Table 1).”
Related to the above, the discussion section is very short. Above several aspects now not included and thus not discussed should be incorporated.
Answer: As requested, the discussion was broadened and deepened in the aspects we considered most significant

Reviewer 2 Report
This manuscript reports a case of desminopathy with a novel bi-allelic variant associated with fatal DCM and a review of the literature around bi-allelic desminopathies. The manuscript is well written however the novelty/significance is not clearly evident, particularly as the variant itself is listed both in Gnomad and in Clinvar.
Comments:
1. The title suggests that the case reported presented only with DCM which is not the case, I would suggest including the early distal muscle weakness in the title as well as the DCM.
2. Please describe the variant in full according to HGVS nomenclature (and also include the NM/transcript information).
3. Please add the ACMG pathogenic scoring for the variant.
4. The same variant has actually been listed twice in Clinvar - yet nothing is mentioned in the manuscript - and the variant is described as "novel". Furthermore the variant is also present in the Gnomad database (although at a very low frequency) and again no mention is made of this. Please include this information in the manuscript.
5. Whilst the variants listed in Clinvar may not have been bi-allelic, the variant is not "novel" but rather it has not been described in the literature. Of interest and for further discussion in the manuscript is whether the Clinvar cases - one listed as likely pathogenic - are mono-allelic and if so why the parents are unaffected in the reported case. Were the parents assessed for any signs of muscle weakness/cardiac features?
4.Related to the point above did the investigators look at levels of DES in the proband and the carrier parents - this would be interesting to see if the parents have any decrease in levels.
Author Response
Response to Reviewers
We thank the reviewers for their comments and suggestions that we have carefully considered in the reviewing of our manuscript. Our point- to- point reply to each reviewer is listed below.
Answer to Reviewer 2
This manuscript reports a case of desminopathy with a novel bi-allelic variant associated with fatal DCM and a review of the literature around bi-allelic desminopathies. The manuscript is well written however the novelty/significance is not clearly evident, particularly as the variant itself is listed both in Gnomad and in Clinvar.
Comments:
- The title suggests that the case reported presented only with DCM which is not the case, I would suggest including the early distal muscle weakness in the title as well as the DCM.
Answer: According to the suggestion, the title of the manuscript was changed as follows: A bi-allelic homozygous variant in DES gene causing early onset distal myopathy and fatal dilated cardiomyopathy. A case report and review of literature.
- Please describe the variant in full according to HGVS nomenclature (and also include the NM/transcript information).
Answer: We added the description of the variant according to HGVS nomenclature [NM_001927.3: c.634C>T; p. Arg212*].
- Please add the ACMG pathogenic scoring for the variant.
Answer: We added the ACMG pathogenic scoring as follows: “The identified DES variant is predicted to be pathogenic according to the ACMG/AMP guidelines (PVS1 very strong, PP5 strong, PM2 supporting), using the default settings of Varsome in April 2022. “
- The same variant has actually been listed twice in Clinvar - yet nothing is mentioned in the manuscript - and the variant is described as "novel". Furthermore, the variant is also present in the Gnomad database (although at a very low frequency) and again no mention is made of this. Please include this information in the manuscript.
Answer: We are grateful for this information that we have included in the manuscript as suggested: “The variant is listed in gnomAD although at a very low frequency (f= 0.00001061), and never in homozygosity. The variant is also listed in ClinVar. A recent update has lifted the clinical significance of this variant to “likely pathogenic” suggesting its possible link to desminopathies (VCV000201722.8).”(pag.
- Whilst the variants listed in Clinvar may not have been bi-allelic, the variant is not "novel" but rather it has not been described in the literature. Of interest and for further discussion in the manuscript is whether the Clinvar cases - one listed as likely pathogenic - are mono-allelic and if so why the parents are unaffected in the reported case. Were the parents assessed for any signs of muscle weakness/cardiac features?
Answer: We deleted the term “novel” in the title to avoid any misleading. We clinically examined the proband's father and sister, both carriers of the same variant, who are healthy and show no signs of muscle weakness or cardiac features. The mother - first cousin of the patient’s father – was not examined as she had died before the patient came to our observation.
- Related to the point above did the investigators look at levels of DES in the proband and the carrier parents - this would be interesting to see if the parents have any decrease in levels.
Answer: DES levels were not studied in the proband, nor in other family members, due to the paucity of muscle sample in the proband, and the father's and sister's refusal to undergo muscle biopsy. However, the variant is expected to cause a nonsense mediated decay and, as a consequence, a strongly reduced protein expression encoded by the mutated alleles.

Round 2
Reviewer 2 Report
I appreciate the updated changes in response to the initial round of review. However I still question the novelty of a case report for a well established DCM gene.